

# Technical Note: Two types of absolute dynamic ocean topography

Peter C. Chu

Naval Ocean Analysis and Prediction Laboratory, Department of Oceanography
Naval Postgraduate School, Monterey, CA 93943, USA

Correspondence to: Peter C. Chu (pcchu@nps.edu)

**Abstract.** Two types of marine geoid and the associated *absolute dynamic ocean topography* (referred as DOT) are presented. The first type is the *average level of sea surface height (SSH) if the* ***water is at rest*** (classical definition). The second type is determined by satellite observation under the condition that usually ***the water is not at rest***. Its mean DOT (MDOT) is comparable to the first type DOT. Respective differences

between the two geoids are that they exclude (include) the gravity anomaly and are non-measurable (measurable) in the first (second) type marine geoid. The first type DOT is determined by a physical principle that the geostrophic balance takes the minimum energy state. On the base of that, a new elliptic equation is derived for the first type DOT. Continuation of geoid from land to ocean leads to an inhomogeneous Dirichlet boundary

condition with the boundary values taking satellite observed second-type MDOT. This well-posed elliptic equation is integrated numerically on 1° grids for the world oceans with the forcing function computed from the World Ocean Atlas (*T*, *S*) fields and the sea-floor topography obtained from the NOAA's ETOPO5 model. Between the first type DOT and second type MDOT, the relative root-mean square (RMS) difference (versus

RMS of the first type DOT) is 38.6% and the RMS difference of the horizontal gradients (versus RMS of the horizontal gradient of the first type DOT) is near 100%.  The standard deviation of horizontal gradients of DOT is nearly twice larger in the second



type (satellite determined marine geoid with gravity anomaly) than in the first type

(geostrophic balance without gravity anomaly). Such difference needs further attention

from oceanographic and geodetic communities, especially the oceanographic

representation of the horizontal gradients of the second type MDOT.

## 1. Introduction

Let a spherical harmonic reference model (flat-Earth approximation) be used with the

coordinates $(x, y, z)$ in zonal, latitudinal, and vertical directions for the gravity computation.

The *absolute dynamic ocean topography* (hereafter referred as DOT) $\hat{D}$, is the sea surface

height (SSH) (waves and tides filtered out) relative to the marine geoid (i.e., the

equipotential surface),

$$\hat{D} = S - \hat{N} , \qquad (1)$$

where $S$ is the SSH; $\hat{N}$ is the marine geoid height above to the reference ellipsoid (Fig.

1), respectively. $\hat{D}$ is an important signal in oceanography; and $\hat{N}$ is of prime interest in

geodesy. The geoid height $\hat{N}(x, y)$ and other associated measurable quantities such as

gravity anomaly $\Delta g(x, y)$ are related to the gravitational potential $V(x, y, z)$ to a first

approximation by Brun's formula,

$$\hat{N}(x, y) = \frac{V(x, y, 0)}{g} , \qquad (2)$$

where $g = 9.81$ m/s$^2$, is the globally mean normal gravity, which is usually represented by

$g_0$ in geodesy. The gravity anomaly is the vertical derivative of the potential

$$\Delta g(x, y) = -\frac{\partial V(x, y, 0)}{\partial z} , \qquad (3)$$

where the gravity potential $V$ satisfies the Laplace equation



$$\frac{\partial^2 V}{\partial x^2} + \frac{\partial^2 V}{\partial y^2} + \frac{\partial^2 V}{\partial z^2} = 0 \ . \tag{4}$$

The vertical deflection is the slope of the geoid

$$\frac{\partial \hat{N}}{\partial x} = \frac{1}{g}\frac{\partial V}{\partial x}, \quad \frac{\partial \hat{N}}{\partial y} = \frac{1}{g}\frac{\partial V}{\partial y} \tag{5}$$

which connects to the gravity anomaly by

$$\frac{\partial (\Delta g)}{\partial z} = g\left(\frac{\partial^2 \hat{N}}{\partial x^2} + \frac{\partial^2 \hat{N}}{\partial y^2}\right) \tag{6}$$

Eq(6) links the vertical gravity gradient to the horizontal Laplacian of the marine geoid

height $\hat{N}$ and serves as the basic principle in the satellite marine geodesy. Since $\hat{D}$ is the

difference of the two large fields $S$ and $\hat{N}$ (two orders of magnitude larger than $\hat{D}$), it is

extremely sensitive to any error in either $S$ or $\hat{N}$ – even 1% error in either field can lead

to error in $\hat{D}$ that is of the same order of magnitudes as $\hat{D}$ itself [Wunsch and Gaposchkin,

1980; Bingham et al., 2008].

60        It has been a long history to observe $S$. The (absolute) DOT ($\hat{D}$) is distinguished

from the relative dynamic ocean topography ($\hat{D}_{rel}$),

$$\hat{D}_{rel} = S - H_r \ . \tag{7}$$

which is the SSH relative to a certain reference depth ($H_r$). Tapley et al. [2003] computed

$\hat{D}_{rel}$ with a reference depth $H_r$ (4000 m or 3000 m if the water depth is less than 4000 m at

the grid). However, (absolute) DOT ($\hat{D}$) [Eq(1)] and relative dynamic ocean topography

($\hat{D}_{rel}$) [Eq(7)] are different. Discussion on difference between $\hat{D}$ and $\hat{D}_{rel}$ is beyond the

scope of this paper.





Before satellite came into practice, $S$ was measured from sparse surveying ships and tide gauge stations located along irregular local coastline. However, $\hat{N}$ was not easy

to observe. Without satellite measurements, the marine geoid is defined as the ***average level of SSH if the water is at rest*** and denoted here by $N$, which is called the classical marine geoid (or first type marine geoid) (Fig. 1a).

The first type marine geoid can be taken as *a standalone concept in oceanography* since it is on the base of the hypothesis (mean SSH when the water at rest) without using

the gravity anomaly. In this framework, the ocean is geostrophically balanced

$$u_g = -\frac{1}{f\hat{\rho}}\frac{\partial \hat{p}}{\partial y}, \quad v_g = \frac{1}{f\hat{\rho}}\frac{\partial \hat{p}}{\partial x}, \tag{8}$$

and hydrostatically balanced,

$$\frac{\partial \hat{p}}{\partial z} = -\hat{\rho}g, \tag{9}$$

for large-scale (i.e., scale > 100 km) processes. Here $(u_g, v_g)$ are geostrophic current

components; $f$ is the Coriolis parameter; $(\hat{p}, \hat{\rho})$ are in-situ pressure and density, respectively, which can be decomposed into

$$\hat{\rho} = \rho_0 + \overline{\rho}(z) + \rho, \quad \hat{p} = -\rho_0 gz + \overline{p}(z) + p. \tag{10}$$

Here, $\rho_0 = 1025 \text{ kg/m}^3$, is the characteristic density; $(\overline{\rho}, \overline{p})$ are horizontally uniform with $\overline{\rho}$ vertically increasing with depth (stable stratification)

$$\partial \overline{\rho} / \partial z \equiv -\rho_0 [n(z)]^2 / g, \tag{11}$$

where $n(z)$ is the buoyancy frequency (or called the Brunt-Vaisala frequency); $(p, \rho)$ are anomalies of pressure and density. Near the ocean surface, it is common to use the





characteristic density and corresponding pressure ( $p_0, \rho_0$ ) to represent ( $\hat{p}, \hat{\rho}$ ). Vertical

integration of (9) from $N$ to $S$ after replacing ( $\hat{p}, \hat{\rho}$ ) by ( $p_0, \rho_0$ ) in (8) and (9) leads to

$$u_g(S) - u_g(N) = -\frac{g}{f}\frac{\partial D}{\partial y}, \quad v_g(S) - v_g(N) = \frac{g}{f}\frac{\partial D}{\partial x}, \quad (12)$$

where

$$D = S - N, \quad (13)$$

 is the first type  DOT. Since the first type marine geoid ($N$) is defined as the *average level*

*of SSH **if the water is at rest***,

$$u_g(N) = 0, \quad v_g(N) = 0, \quad (14)$$

the horizontal gradient of $D$ represents the absolute surface geostrophic currents.

After satellites came into practice, SSH has been observed with high precision and

unique resolution with altimetry above a reference ellipsoid (not geoid) (Fu and Haines

2013). Two Gravity Recovery and Climate Experiment (GRACE) satellites, launched in

2002, provide data to compute the marine geoid [called the GRACE Gravity Model

(GGM)] (see website: http://www.csr.utexas.edu/grace/) [Tapley et al., 2003; Shum et al.,

2011]. This marine geoid is the solution of Eq(6),

$$\frac{\partial^2 N_*}{\partial x^2} + \frac{\partial^2 N_*}{\partial y^2} = \frac{1}{g}\frac{\partial(\Delta g)}{\partial z}$$

where $N_*$ is the  satellite  determined marine geoid from the measurable gravity anomaly

$\Delta g$ , and called  the second type marine geoid (Fig. 1b), which is different from $N$, defined

by (14). Correspondingly, the second type DOT is defined by

$$D_* = S - N_*(t), \quad (15)$$





where $N_*(t)$ changes with time due to temporally varying gravity anomaly $\Delta g$. Thus, comparison between the first-type and second-type geoids should be conducted between $N$

and $\bar{N}_*$. Here, $\bar{N}_*$ is the temporally mean of $N_*(t)$. As for DOT, the first type DOT ($D$) should be compared to the second type mean DOT (MDOT),

$$\bar{D}_* = S - \bar{N}_* \qquad (16)$$

The second type marine geoid can be taken as *a standalone concept in marine geodesy* since it is on the base of the equipotential surface using the gravity anomaly.

Melnichenko et al. [2010] produced high-resolution mean (1993-2002) second type MDOT (i.e., $\bar{D}_*$) using the combined data of surface drifters, satellite altimetry, NCEP reanalysis wind, and GRACE satellite mission-based product and distributed at the website: http://apdrc.soest.hawaii.edu/projects/DOT/.

A question arises: Are the two types of DOT ($D$, $\bar{D}_*$) or marine geoid ($N$, $\bar{N}_*$) the

same? This paper will answer the question using the temporally averaged SSH and marine geoid from NASA's satellite altimetric and gravimetric measurements [i.e., the second type MDOT ($\bar{D}_*$)], and solving a new elliptic equation of $D$ numerically. Given ($S$, $\bar{N}_*$, $D$) leads to the answer of the question.

Rest of the paper is outlined as follows. Section 2 describes the change of DOT due

to the change of marine geoid from first to second type. Section 3 describes geostrophic currents and energy related to the first type DOT. Section 4 presents the governing equation of the first type DOT with the boundary condition at the coasts. Section 5 shows the numerical solution for the world oceans. Section 6 evaluates the change of global DOT from first to second type with oceanographic implication. Section 7 concludes the studies.



## 2. Change of DOT from first to second type

The second type MDOT ($\bar{D}_*$) data were downloaded from the NASA/JPL website:

https://grace.jpl.nasa.gov/data/get-data/dynamic-ocean-typography/. This dataset was

subtraction of a  second type marine geoid of GRACE [Bingham et al, 2011] from a mean

(1993 to 2006) altimetric sea surface. Change of marine geoid from first type ($N$) to second

type ($\bar{N}_*$) is represented by

$$\Delta N = \bar{N}_* - N \qquad (17)$$

Correspondingly, change of DOT is given by

$$\Delta D = \bar{D}_* - D = -\Delta N \qquad (18)$$

where (13) and (16) are used. $\Delta D$ is of interest in oceanography. $\Delta N$ is of interest in

geodesy. Eq(18) shows that the key issue to evaluate $\Delta D$ is to determine $D$ (i.e., first type

DOT).

When the frictional force is negligible, the potential vorticity (PV) is conserved.

The geostrophic current reaches the minimum energy state (Appendix A). On the base of

the minimum energy state, an elliptic partial differential equation for $D$ is derived with

coefficients containing sea-floor topography $H$, and forcing function containing

temperature and salinity fields.

If $\Delta D$ is negligible in comparison to $D$, change of marine geoid from $N$ to $\bar{N}_*$ does

not change absolute DOT's oceanographic interpretation, i.e., the horizontal gradient of

$\bar{D}_*$ also represents the absolute surface geostrophic currents. If $\Delta D$ is not negligible, the

horizontal gradient of $\bar{D}_*$ does not represent the absolute surface geostrophic currents.

## 3. Geostrophic currents and energy





Eq.(10) implies,

$$\frac{\partial \hat{\rho}}{\partial x} = \frac{\partial \rho}{\partial x}, \quad \frac{\partial \hat{\rho}}{\partial y} = \frac{\partial \rho}{\partial y}, \tag{19}$$

$$\frac{\partial \hat{p}}{\partial x} = \frac{\partial p}{\partial x}, \quad \frac{\partial \hat{p}}{\partial y} = \frac{\partial p}{\partial y}. \tag{20}$$

Using the first type marine geoid $N$, the horizontal gradient of $D$ leads to the surface

geostrophic currents [see Eqs(12) and (14)]. Integration of the thermal wind relation

$$\frac{\partial u_g}{\partial z} = \frac{g}{f \rho_0} \frac{\partial \rho}{\partial y}, \quad \frac{\partial v_g}{\partial z} = -\frac{g}{f \rho_0} \frac{\partial \rho}{\partial x}, \tag{21}$$

from the ocean surface to depth $z$ leads to depth-dependent geostrophic currents,

$$u_g(z) = u_g(S) + u_{BC}(z), \quad v_g(z) = v_g(S) + v_{BC}(z) \tag{22}$$

where

$$u_{BC}(z) = -\frac{g}{f \rho_0} \int_z^0 \frac{\partial \rho}{\partial y} dz', \quad v_{BC}(z) = \frac{g}{f \rho_0} \int_z^0 \frac{\partial \rho}{\partial x} dz', \tag{23}$$

are the baroclinic geostrophic currents. Here, $f = 2\Omega \sin(\varphi)$ is the Coriolis parameter; $\Omega =$

$2\pi/(86400\ \text{s})$ is the mean Earth rotation rate; $\varphi$ is the latitude.

The volume integrated total energy, i.e., sum of kinetic energy of the geostrophic

currents and the available potential energy [Oort et al., 1989], for an ocean basin ($W$) is

given by

$$E = \iiint_W \left[ \frac{1}{2}(u_g^2 + v_g^2) + \frac{g^2 \rho^2}{2 \rho_0^2 n^2} \right] dxdydz. \tag{24}$$

Substitution of (22) and (23) into (24) leads to

$$E(D_x, D_y, \rho) = \frac{g^2}{2} \iiint_W \left[ (-D_y + \frac{f u_{BC}}{g})^2 / f^2 + (D_x + \frac{f v_{BC}}{g})^2 / f^2 + \frac{\rho^2}{\rho_0^2 n^2} \right] dxdydz$$



$$= \frac{g^2}{2} \iiint_W \left[ D_x^2 / f^2 + D_y^2 / f^2 + 2D_x \frac{v_{BC}}{fg} - 2D_y \frac{u_{BC}}{fg} \right] dxdydz$$

$$+ \frac{1}{2} \iiint_W \left[ u_{BC}^2 + v_{BC}^2 + \frac{g^2 \rho^2}{\rho_0^2 n^2} \right] dxdydz \tag{25}$$


## 4. Governing equation of $D$

For a given density field, the second integration in the right side of (25) is known. The geostrophic currents taking the minimum energy state provides a constraint for $D$,

$$G(D_x, D_y) \equiv \iiint_W \left[ \left( D_x^2 + D_y^2 + 2D_x \frac{fv_{BC}}{g} - 2D_y \frac{fu_{BC}}{g} \right) / f^2 \right] dxdydz \to \min. \tag{26}$$

The three-dimensional integration (26) over the ocean basin is conducted by

$$\iiint_W [...] dxdydz = \iint_R \left\{ \int_{-H}^0 [...] dz \right\} dxdy \tag{27}$$

where $R$ is the horizontal area of the water volume, $H$ is the water depth. Thus, Eq(26)

becomes

$$G(D_x, D_y) = \iint_R L(D_x, D_y) dxdy \to \min, \tag{28}$$

$$L(D_x, D_y) \equiv \left[ H(D_x^2 + D_y^2) + 2D_x Y - 2D_y X \right] / f^2 \tag{29}$$

where the parameters $(X, Y)$ are given by

$$X(x,y) \equiv \frac{f}{g} \int_{-H}^0 u_{BC} dz = -\frac{1}{\rho_0} \int_{-H}^0 \int_z^0 \frac{\partial \hat{\rho}}{\partial y} dz' dz \tag{30}$$

$$Y(x,y) \equiv \frac{f}{g} \int_{-H}^0 v_{BC} dz = \frac{1}{\rho_0} \int_{-H}^0 \int_z^0 \frac{\partial \hat{\rho}}{\partial x} dz' dz, \tag{31}$$

which represent vertically integrated baroclinic geostrophic currents scaled by the factor

$f/g$ (unit: m). Here, Eq.(19) is used (i.e., horizontal gradient of in-situ density is the same

as that of density anomaly).





The Euler-Lagragian equation of the functional (28) is given by

$$\frac{\partial L}{\partial D} - \frac{\partial}{\partial x}\left(\frac{\partial L}{\partial D_x}\right) - \frac{\partial}{\partial y}\left(\frac{\partial L}{\partial D_y}\right) = 0. \tag{32}$$

Substitution of (29) into (32) gives an elliptic partial differential equation (i.e., the

governing equation) for the first type DOT (i.e., $D$),

$$f^2 \nabla\left[\left(H/f^2\right)\nabla D\right] = -F,$$

or

$$H\left[\nabla^2 D + r^{(x)}\frac{\partial D}{\partial x} + r^{(y)}\frac{\partial D}{\partial y} - 2(\beta/f)\frac{\partial D}{\partial y}\right] = -F, \tag{33}$$

where

$$F \equiv \left(\frac{\partial Y}{\partial x} - \frac{\partial X}{\partial y}\right), \quad \nabla \equiv \mathbf{i}\frac{\partial}{\partial x} + \mathbf{j}\frac{\partial}{\partial y} \tag{34}$$

$$r^{(x)} \equiv \frac{1}{H}\frac{\partial H}{\partial x}, \quad r^{(y)} \equiv \frac{1}{H}\frac{\partial H}{\partial y}, \quad \beta = \frac{2\Omega}{a}\cos(\varphi), \tag{35}$$

where $a = 6{,}370$ km, is the mean earth radius. The geostrophic balance does not exist at the

equator. The Coriolis parameter $f$ needs some special treatment for low latitudes. In this

study, $f$ is taken as $2\Omega\sin(5\pi/180)$ if latitude between 10ºN to 0º; and as

$-2\Omega\sin(5\pi/180)$ if latitude between 0º to 10ºS.

Let $\Gamma$ be the coastline of ocean basin. Continuation of geoid from land to oceans

gives

$$N\big|_\Gamma = N_l\big|_\Gamma, \quad \bar{N}_*\big|_\Gamma = N_l\big|_\Gamma, \tag{36}$$

which leads to

$$N\big|_\Gamma = \bar{N}_*\big|_\Gamma. \tag{37}$$



Here, $N_l$ is the geoid over land. The boundary condition (37) can be rewritten as

$$D\big|_\Gamma = (S-N)\big|_\Gamma = (S-\bar{N}_*)\big|_\Gamma = \bar{D}_*\big|_\Gamma \tag{38}$$

which is boundary condition of $D$.

## 5. Numerical solution of $D$

The well-posed elliptic equation (33) is integrated numerically on $1^\circ \times 1^\circ$ grids for the world

oceans with the boundary values [i.e., (38)] taken from the MDOT (1993-2006) field (i.e.,

$\bar{D}_*$), at the NASA/JPL website: https://grace.jpl.nasa.gov/data/get-data/dynamic-ocean-

typography/ ($0.5^\circ$ interpolated into $1^\circ$ resolution). The forcing function $F$ is calculated on

$1^\circ \times 1^\circ$ grid from the World Ocean Atlas 2013 (WOA13) temperature and salinity fields,

which was downloaded from the NOAA National Centers for Environmental Information

(NCEI)   website: https://www.nodc.noaa.gov/OC5/woa13/woa13data.html. The three

dimensional density was calculated using the international thermodynamic equation of

seawater   -2010,   which   is   downloaded   from   the   website:

http://unesdoc.unesco.org/images/0018/001881/188170e.pdf.   The   ocean   bottom

topography data $H$ was downloaded from the NECI 5-Minute Gridded Global Relief Data

Collection   at   the   website:   https://www.ngdc.noaa.gov/mgg/fliers/93mgg01.html.

Discretization of the elliptic equation (33) and numerical integration are given in Appendix

B.

### 6. Difference between the Two DOTs

The first type global DOT ($D_{ij}$) (Fig. 2a) is the numerical solution of the elliptic equation

(33) with the boundary condition (38). The second type global MDOT ($\bar{D}_{*ij}$) (Fig. 2b) is





downloaded from the NASA/JPL website: https://grace.jpl.nasa.gov/data/get-data/dynamic-ocean-typography/. Difference between the two DOTs,

$$\Delta D_{ij} = \bar{D}_{*ij} - D_{ij}, \tag{39}$$

is evident in the world oceans (Fig. 2c). Here, $(i, j)$ denote the horizontal grid point. The relative root-mean-square (RMS) of $\Delta D$ is given by

$$\text{RRMS}(\Delta D) = \frac{\sqrt{\dfrac{1}{M}\sum_i \sum_j (\Delta D_{ij})^2}}{\sqrt{\dfrac{1}{M}\sum_i \sum_j (D_{ij})^2}} = 0.386. \tag{40}$$

where M = 38,877 is the number of total grid points. Both $D$ and $\bar{D}_*$ have positive and negative values. The arithmetic mean values (0.524 cm, -3.84 cm) are much smaller than

the RMS mean values. They are an order of magnitude smaller than the corresponding standard deviations (54.9 cm, 71.2 cm) (see Figs. 2d and 2e). The magnitudes of $D$ and $\bar{D}_*$ are represented by their root-mean squares, which are close to their standard deviations.

Histograms of for $D_{ij}$ (Fig. 2d) and $\bar{D}_{*ij}$ (Fig. 2e) are both non-Gaussian and negatively skewed. The major difference between the two is the single modal for $D_{ij}$ with a peak at

around 20 cm and the bi-modal for $\bar{D}_{*ij}$ with a high peak at around 30 cm and a low peak at -140 cm. The statistical parameters are different, such as mean value and standard deviation are (0.524 cm, 54.9 cm) for $D_{ij}$, and (-3.84 cm, 71.2 cm) for $\bar{D}_{*ij}$. Skewness and kurtosis are (-0.83, 3.01) for $D_{ij}$, and (-0.87, 2.80) for $\bar{D}_{*ij}$.

Horizontal gradients of the DOT, $(\partial D_{ij}/\partial x,\ \partial D_{ij}/\partial y)$ and $(\partial \bar{D}_{*ij}/\partial x, \partial \bar{D}_{*ij}/\partial y)$, have

oceanographic significance (i.e., related to the geostrophic currents). They are calculated



using the central difference scheme at inside-domain grid points and the first order forward/backward difference scheme at grid points next to the boundary. Difference in global $\partial D_{ij}/\partial x$ (Fig. 3a) and $\partial \bar{D}_{*ij}/\partial x$ (Fig. 3b) is evident with much smaller-scale structures in $\partial \bar{D}_{*ij}/\partial x$. The difference between the two gradients (Fig. 3c),

$$\Delta(\partial D_{ij}/\partial x) = \partial \bar{D}_{*ij}/\partial x - \partial D_{ij}/\partial x \qquad (41)$$

has the same order of magnitudes as the gradients themselves with the relative root-mean-square (RMS) of $\Delta(\partial D/\partial x)$,

$$\mathrm{RRMS}\left[\Delta(\partial D/\partial x)\right] = \frac{\sqrt{\dfrac{1}{M}\sum_i\sum_j\left[\Delta(\partial D_{ij}/\partial x)\right]^2}}{\sqrt{\dfrac{1}{M}\sum_i\sum_j(\partial D_{ij}/\partial x)^2}} = 1.04, \qquad (42)$$

which implies that the non-surface latitudinal geostrophic current component of the second type MDOT has the same order of magnitude as the surface latitudinal geostrophic current component of the first type DOT. Histograms of for $\partial D_{ij}/\partial x$ (Fig. 3d) and $\partial \bar{D}_{*ij}/\partial x$ (Fig. 3e) are near symmetric with mean values around $(-1.29, -0.78)\times10^{-8}$ and standard deviations $(2.69, 4.95)\times10^{-7}$. The standard deviation of $\partial \bar{D}_{*ij}/\partial x$ is almost twice that of $\partial D_{ij}/\partial x$.

Similarly, difference in global $\partial D_{ij}/\partial y$ (Fig. 4a) and $\partial \bar{D}_{*ij}/\partial y$ (Fig. 4b) is evident with much smaller-scale structures in $\partial \bar{D}_{*ij}/\partial y$. The difference between the two gradients (Fig. 4c),

$$\Delta(\partial D_{ij}/\partial y) = \partial \bar{D}_{*ij}/\partial y - \partial D_{ij}/\partial y \qquad (43)$$



has the same order of magnitudes as the gradients themselves with the relative root-mean-

square (RMS) of $\Delta(\partial D / \partial y)$,

$$\text{RRMS}\big[\Delta(\partial D / \partial y)\big] = \frac{\sqrt{\dfrac{1}{M}\sum_i \sum_j \big[\Delta(\partial D_{ij} / \partial y)\big]^2}}{\sqrt{\dfrac{1}{M}\sum_i \sum_j (\partial D_{ij} / \partial y)^2}} = 0.98, \qquad (44)$$

which implies that the non-surface zonal geostrophic current component of the second type

MDOT has the same order of magnitude as the surface zonal geostrophic current

component of the first type DOT.    Histograms of $\partial D_{ij}/\partial y$ (Fig. 4d) and $\partial \bar{D}_{*ij} / \partial y$ (Fig.

4e) are also near symmetric with the mean values around $(2.32, 1.18)\times 10^{-7}$ and standard

deviations $(1.20, 2.44)\times 10^{-6}$.    The standard deviation of $\partial \bar{D}_{*ij} / \partial y$ is almost twice that of

$\partial D_{ij}/\partial y$. The denominators of (42) and (44) represent the magnitudes of the horizontal

gradients of the first type DOT.

## 7. Conclusions

Change of marine geoid from classical defined (first type) to satellite determined (second

type, *standalone concept in marine geodesy*) largely affects oceanography. With the

classical defined marine geoid (***average level of SSH if the water is at rest***) the first type

DOT represents the absolute geostrophic currents at the surface. With the satellite

determined (second type) marine geoid by Eq(6), the second type MDOT might not

represent the absolute geostrophic currents at the surface. The difference between the two

types of DOT represents the component in addition to the absolute geostrophic currents at

the surface.

With conservation of potential vorticity, geostrophic balance represents the

minimum energy state in an ocean basin where the mechanical energy is conserved. A new

governing elliptic equation of first type DOT is derived with water depth ($H$) in the coefficients and the three dimensional temperature and salinity in the forcing function. This governing elliptic equation is well posed. Continuation of geoid from land to ocean leads to an inhomogeneous Dirichlet boundary condition.

        Difference between the two types of absolute DOT is evident with relative root-
mean-square difference of 38.6%. Horizontal gradients (representing geostrophic currents) of the two type DOTs are different with much smaller-scale structures in the second type absolute DOT. Relative root-mean-square difference is near 1.0 in both ($x$, $y$) components of the DOT gradient, which implies that the non-absolute surface geostrophic currents identified from the second type has the same order of magnitudes of the absolute surface
geostrophic currents identified by the first type absolute DOT.

        The notable difference between the two types of absolute DOT raises more questions in oceanography and marine geodesy: Is there any theoretical foundation to connect the classical marine geoid (*standalone concept in oceanography* using the principle of surface geostrophic currents without $\Delta g$ ) to the satellite determined marine
geoid (*standalone concept in marine geodesy* using $\Delta g$ without the principle of surface geostrophic currents)? How can the satellite determined marine geoid using the gravity anomaly ($\Delta g$ ) be conformed to the basic physical oceanography principle of surface geostrophic currents? What is the interpretation of the horizontal gradients of the second type MDOT ($\bar{D}_*$)? Is there any evidence or theory to show $[u_g(\bar{N}_*) = 0, v_g(\bar{N}_*) = 0]$
similar to Eq.(14)? More observational and theoretical studies are needed in order to solve those problems. The main challenge for oceanography is how to use the satellite altimetry observed SSH and subtract the satellite gravimetry or gradiometry determined gravity field

(with gravity anomaly $\Delta g$) to infer the ocean general circulations at the surface. A new

theoretical framework rather than the geostrophic constraint needs to be established.

The GOCE determined satellite data-only geoid model is more accurate and with

higher resolution than GRACE. Change of GRACE to GOCE geoid model may increase

the accuracy of the calculation of the second type absolute DOT. However, such a

replacement does not solve the fundamental problem presented here, i.e., incompatibility

between satellite determined marine geoid using the gravity anomaly ($\Delta g$) and the

classical marine geoid (mean SSH when the water at rest) on the base of the basic physical

oceanography principle of surface geostrophic currents.

Finally, the mathematical framework described here [i.e., the elliptic equation (33)]

may lead to a new inverse method for calculating three-dimensional absolute geostrophic

velocity from temperature and salinity fields since the surface absolute geostrophic velocity

is the solution of (33). This will be useful in addition to the existing β-spiral method

[Stommel and Schott, 1977], box model [Wunsch, 1978], and P-vector method [Chu, 1995;

Chu et al., 1998, 2000].

***Acknowledgments***. The author thanks Mr. Chenwu Fan for invaluable comments and

computational assistance, NOAA/NCEI for the WOA-2013 (T, S) and ETOPO5 sea-floor

topography data, and NASA/JPL (second type) MDOT data.

## Appendix A. Geostrophic balance as a minimum energy state in an energy conserved basin

In large scale motion (small Rossby number) with the Boussinesq approximation, the

linearized PV ($\Pi$) is given by



$$\Pi \approx [f + (\frac{\partial v}{\partial x} - \frac{\partial u}{\partial y})]\frac{\partial \hat{\rho}}{\partial z} \approx f(-\frac{\rho_0 n^2}{g} + \frac{\partial \rho}{\partial z}) - \frac{\rho_0 n^2}{g}(\frac{\partial v}{\partial x} - \frac{\partial u}{\partial y}).$$
(A1)

where, $\rho_0 = 1025$ kg m$^{-3}$ is the characteristic density. Without the frictional force and zero horizontally integrated buoyancy flux at the surface and bottom, the energy (including kinetic and available potential energies) is conserved in a three dimensional ocean basin $(V)$

$$E = \iiint_V J dx dy dz, \quad J \equiv \frac{1}{2}(u^2 + v^2 + w^2) + \frac{g^2 \rho^2}{2 \rho_0^2 n^2},$$
(A2)

$$\frac{dE}{dt} = 0$$
(A3)

The two terms of $J$ are kinetic energy, and available potential energy. To show the geostrophic balance taking the minimum energy state for a given linear PV [see (A1)], the constraint is incorporated by extremizing the integral (see also in Vallis 1992)

$$I \equiv \iiint_V \left\{ \begin{array}{l} \frac{1}{2}(u^2 + v^2 + w^2) + \frac{g^2 \rho^2}{2 \rho_0^2 n^2} \\ + \mu(x, y, z)\left[f(-\frac{\rho_0 n^2}{g} + \frac{\partial \rho}{\partial z}) - \frac{\rho_0 n^2}{g}(\frac{\partial v}{\partial x} - \frac{\partial u}{\partial y})\right] \end{array} \right\} dx dy dz$$
(A4)

where $\mu(x, y, z)$ is the Lagrange multiplier, which is a function of space. If it were a constant, the integral would merely extremize energy subject to a given integral of PV, and rearrangement of PV would leave the integral unaltered. Extremization of the integral (A4) gives the three Euler-Lagrange equations,

$$\frac{\partial K}{\partial \rho} - \frac{\partial}{\partial z}\frac{\partial K}{\partial \rho_z} = 0,$$
(A5)

$$\frac{\partial K}{\partial u} - \frac{\partial}{\partial y}\frac{\partial K}{\partial u_y} = 0,$$
(A6)





$$\frac{\partial K}{\partial v} - \frac{\partial}{\partial x}\frac{\partial K}{\partial v_x} = 0. \qquad (A7)$$

where $K$ is in the integrand appearing in (A4). Substitution of $K$ into (A5), (A6), (A7) leads to

$$\frac{g^2}{\rho_0^2 n^2}\rho = f\frac{\partial \mu}{\partial z}, \qquad (A8)$$

$$u = \frac{\rho_0 n^2}{g}\frac{\partial \mu}{\partial y}, \quad v = -\frac{\rho_0 n^2}{g}\frac{\partial \mu}{\partial x}. \qquad (A9)$$

Differentiation of (A9) with respect to $z$ and use of (A8) leads to

$$\frac{\partial u}{\partial z} = \frac{g}{f\rho_0}\frac{\partial \rho}{\partial y} = \frac{\partial u_g}{\partial z}, \quad \frac{\partial v}{\partial z} = -\frac{g}{f\rho_0}\frac{\partial \rho}{\partial x} = \frac{\partial v_g}{\partial z}, \qquad (A10)$$

which shows that $(u, v) = (u_g, v_g)$ have the minimum energy state.

**Appendix B. Numerical solution of the equation (33)**

Let the three axes $(x, y, z)$ be discretized into local rectangular grids in horizontal and non-uniform grids in vertical $(x_{i,j}, y_{i,j}, z_k)$ with cell sizes $(1°×1°)$,

$$\Delta y = \frac{\pi}{360}r_E, \ \Delta x_j = \Delta y\cos\phi_j, \ \Delta z_k = z_k - z_{k+1},$$

$$i = 1, 2, ... , I; \ j = 1, 2, ..., J; \ k = 1, 2, ..., K_{i,j} \qquad (B1)$$

where $k = 1$ for the surface, $k = K_{ij}$ for the bottom; $\phi_j$ is the latitude of the grid point; $r_E = 6{,}371$ km, is the earth radius; $I = 360$; $J = 180$. The subscripts in $K_{i,j}$ in (B1) indicates non-uniform water depth in the region.

The parameters $(X_{i,j}, Y_{i,j})$ in (30) and (31) (in Section 4) are calculated by



$$X_{i,j} \equiv \frac{1}{4\rho_0} \sum_{k=2}^{K_{i,j}} \sum_{l=1}^{k} \left[ \left( \frac{\hat{\rho}_{i,j+1,l} - \hat{\rho}_{i,j,l}}{\Delta y} + \frac{\hat{\rho}_{i+1,j+1,l} - \hat{\rho}_{i+1,j,l}}{\Delta y} + \frac{\hat{\rho}_{i,j+1,l+1} - \hat{\rho}_{i,j,l+1}}{\Delta y} + \frac{\hat{\rho}_{i+1,j+1,l+1} - \hat{\rho}_{i+1,j,l+1}}{\Delta y} \right) \Delta z_k \left( \frac{\Delta z_l + \Delta z_{l+1}}{2} \right) \right] \quad (B2)$$


$$Y_{i,j} \equiv \frac{1}{4\rho_0} \sum_{k=2}^{K_{i,j}} \sum_{l=1}^{k} \left[ \left( \frac{\hat{\rho}_{i+1,j,l} - \hat{\rho}_{i,j,l}}{\Delta x_j} + \frac{\hat{\rho}_{i+1,j+1,l} - \hat{\rho}_{i,j+1,l}}{\Delta x_j} + \frac{\hat{\rho}_{i+1,j,l+1} - \hat{\rho}_{i,j,l+1}}{\Delta x_j} + \frac{\hat{\rho}_{i+1,j+1,l+1} - \hat{\rho}_{i,j+1,l+1}}{\Delta x_j} \right) \Delta z_k \left( \frac{\Delta z_l + \Delta z_{l+1}}{2} \right) \right] \quad (B3)$$

which gives the discretized forcing function

$$F_{i,j} = \frac{Y_{i+1,j} - Y_{i-1,j}}{2\Delta x_j} - \frac{X_{i,j+1} - X_{i,j-1}}{2\Delta y} \quad (B4)$$

The governing equation (33) is discretized by

$$\frac{D_{i+1,j} - 2D_{i,j} + D_{i-1,j}}{(\Delta x_j)^2} + \frac{D_{i,j+1} - 2D_{i,j} + D_{i,j-1}}{(\Delta y)^2}$$

$$+ r_{ij}^{(x)} \frac{D_{i+1,j} - D_{i-1,j}}{2\Delta x_j} + \left( r_{ij}^{(y)} - \frac{2\beta_j}{f_j} \right) \frac{D_{i,j+1} - D_{i,j-1}}{2\Delta y} = -\frac{F_{ij}}{H_{ij}} \quad (B5)$$

which is reorganized by

$$2(1+\cos^2\phi_j)D_{i,j} = (1 + \frac{1}{2} r_{ij}^{(x)} \Delta y \cos\phi_j)D_{i+1,j} + (1 - \frac{1}{2} r_{ij}^{(x)} \Delta y \cos\phi_j)D_{i-1,j}$$

$$+ \cos^2\phi_j \left[ 1 + \left( r_{ij}^{(y)} - \frac{2\cot\phi_j}{r_E} \right) \frac{\Delta y}{2} \right] D_{i,j+1} + \cos^2\phi_j \left[ 1 - \left( r_{ij}^{(y)} - \frac{2\cot\phi_j}{r_E} \right) \frac{\Delta y}{2} \right] D_{i,j-1} \quad (B6)$$

$$+ \frac{F_{ij}}{H_{ij}} (\Delta y)^2 \cos^2\phi_j$$

The iteration method is used to solve the algebraic equation (B6) with large value of $I \times J$.

It starts from the 0-step,

$$D_{ij}^{(0)} = 0, \quad i = 1,\ 2,\ \dots, I; \ j = 1,\ 2,\ \dots, J \quad (B7)$$




With the given boundary condition (38) (see Section 4) and forcing function (B4), the first

type DOT at the grid points can be computed from steps $n$ to $n+1$,

$$2(1+\cos^2\phi_j)D_{ij}^{(n+1)} = (1+\frac{1}{2}r_{ij}^{(x)}\Delta y\cos\phi_j)D_{i+1,j}^{(n)} + (1-\frac{1}{2}r_{ij}^{(x)}\Delta y\cos\phi_j)D_{i-1,j}^{(n)}$$

$$+\cos^2\phi_j\left[1+\left(r_{ij}^{(y)}-\frac{2\cot\phi_j}{r_E}\right)\frac{\Delta y}{2}\right]D_{i,j+1}^{(n)} + \cos^2\phi_j\left[1-\left(r_{ij}^{(y)}-\frac{2\cot\phi_j}{r_E}\right)\frac{\Delta y}{2}\right]D_{i,j-1}^{(n)} + \frac{F_{ij}}{H_{ij}}(\Delta y)^2\cos^2\phi_j$$

(B8)

Such iteration continues until the relative root-mean square difference reaching the

criterion,

$$r = \frac{\sqrt{\frac{1}{M}\sum_{i=1}^{I}\sum_{j=1}^{J}\left[D_{ij}^{(n+1)}-D_{ij}^{(n)}\right]^2}}{\sqrt{\frac{1}{M}\sum_{i=1}^{I}\sum_{j=1}^{J}\left[D_{ij}^{(n)}\right]^2}} < 10^{-6},$$

(B9)

where $M = 38,877$, is the total number of the grid points on the ocean surface.

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

**Figure Captions**

Figure 1. Two types of marine geoid and DOT: (a) first type with $N$ the average level of SSH if water at rest (classical definition), and (b) second type with satellite determined $N_*$ (water not at rest).

Figure 2. (a) First type DOT (i.e., $D$) which is the solution of (33) with boundary condition of (38) (unit: cm), (b) second type MDOT (1993-2006) (i.e., $\bar{D}_*$) (unit: cm) downloaded from the NASA/JPL website: https://grace.jpl.nasa.gov/data/get-data/dynamic-ocean-typography, (c) difference between the two DOTs (i.e., $\Delta D$), (d) histogram of global $D$, and (e) histogram of global $\bar{D}_*$.




Figure 3. Derivatives in the *x*-direction of (a) the first type DOT (i.e., $\partial D/\partial x$), (b) the second MDOT (i.e., $\partial \bar{D}_* / \partial x$), (c) the difference $\Delta(\partial D / \partial x) = \partial \bar{D}_* / \partial x - \partial D / \partial x$, (d) histogram of global $\partial D/\partial x$, and (e) histogram of global $\partial \bar{D}_* / \partial x$.

Figure 4. Derivatives in the *y*-direction of (a) the first type DOT (i.e., $\partial D/\partial y$), (b) the second type MDOT (i.e., $\partial \bar{D}_* / \partial y$), and (c) the difference $\Delta(\partial D / \partial y) = \partial \bar{D}_* / \partial y - \partial D / \partial y$, (d) histogram of global $\partial D/\partial y$, and (e) histogram of global $\partial \bar{D}_* / \partial y$.

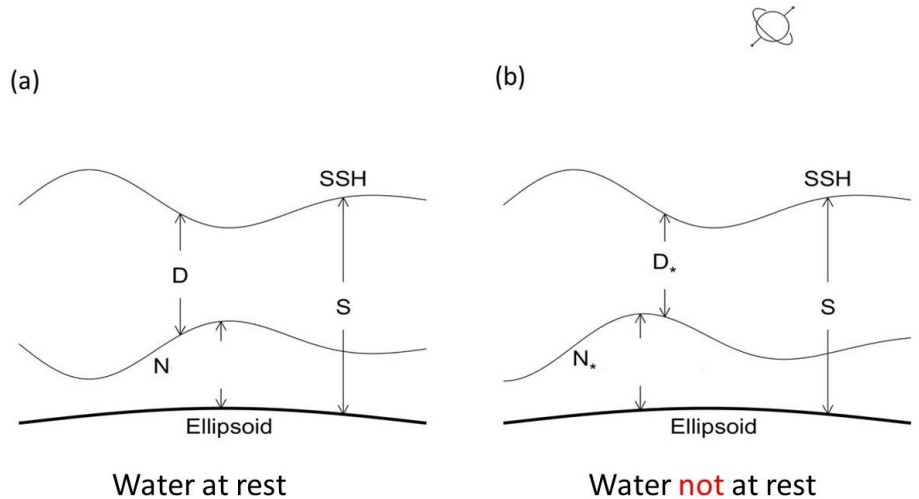

Figure 1. Two types of marine geoid and DOT: (a) first type with *N* the average level of SSH if water at rest (classical definition), and (b) second type with satellite determined *N*∗ (water not at rest).




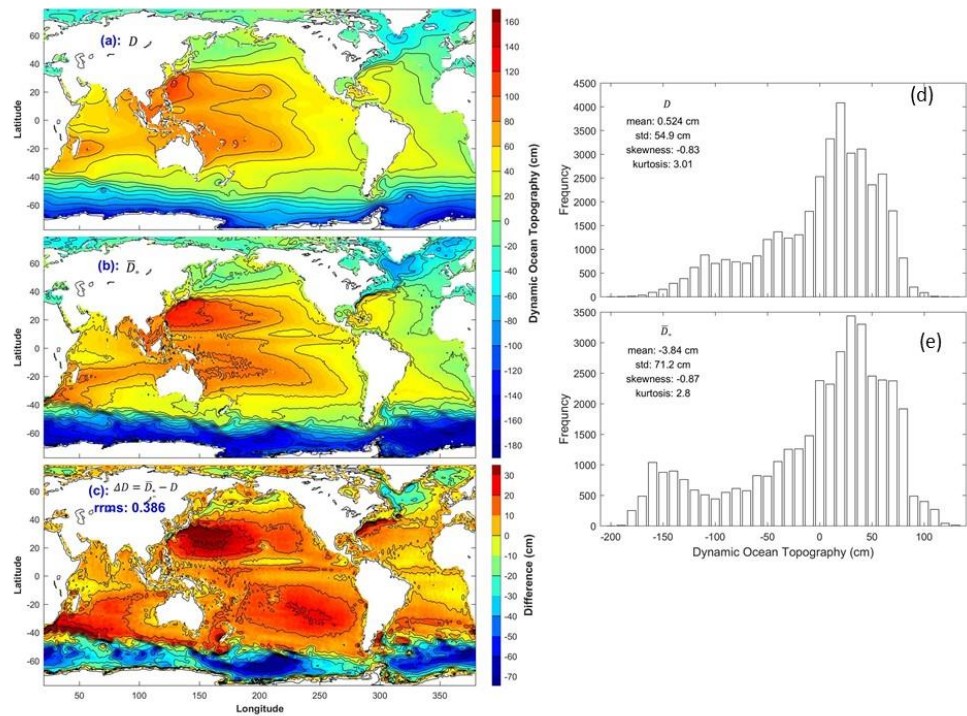

Figure 2. (a) First type DOT (i.e., $D$) which is the solution of (33) with boundary condition of (38) (unit: cm), (b) second type MDOT (1993-2006) (i.e., $\bar{D}_*$) (unit: cm) downloaded from the NASA/JPL website: https://grace.jpl.nasa.gov/data/get-data/dynamic-ocean-typography, (c) difference between the two DOTs (i.e., $\Delta D$), (d) histogram of global $D$, and (e) histogram of global $\bar{D}_*$.






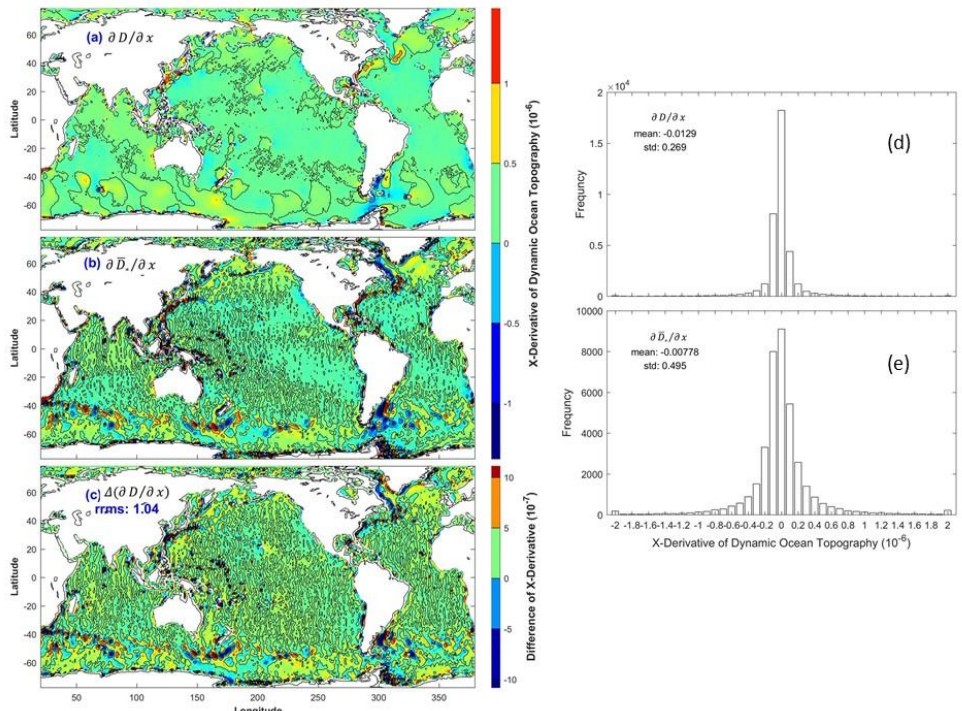

Figure 3. Derivatives in the $x$-direction of (a) the first type DOT (i.e., $\partial D/\partial x$), (b) the second MDOT (i.e., $\partial \bar{D}_* / \partial x$), (c) the difference $\Delta(\partial D / \partial x) = \partial \bar{D}_* / \partial x - \partial D / \partial x$, (d) histogram of global $\partial D/\partial x$, and (e) histogram of global $\partial \bar{D}_* / \partial x$.




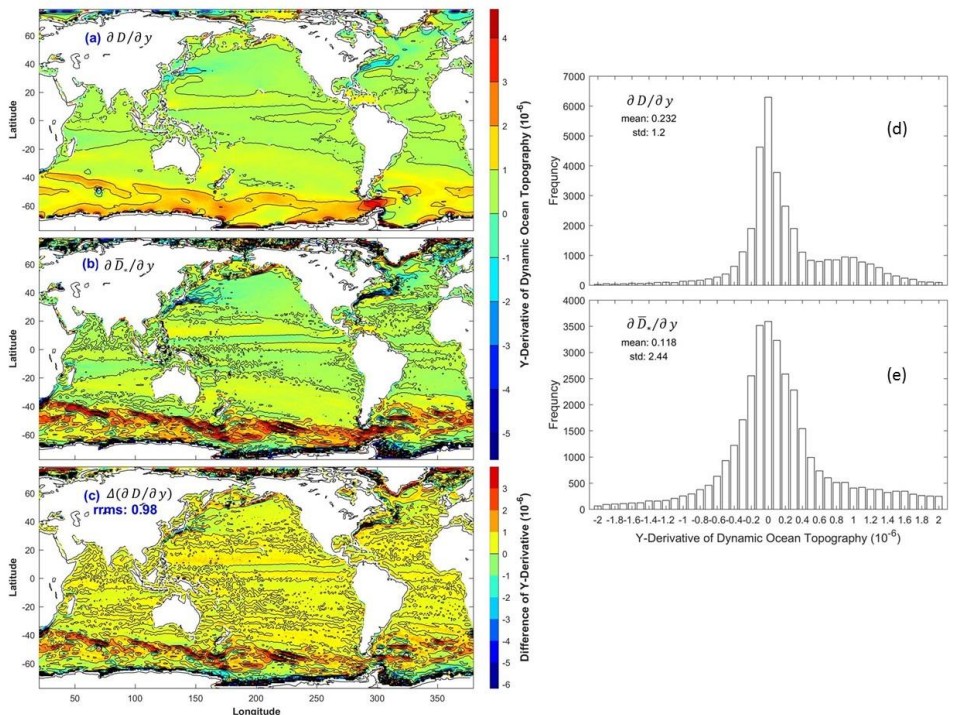

Figure 4. Derivatives in the *y*-direction of (a) the first type DOT (i.e., $\partial D/\partial y$), (b) the second

type MDOT (i.e., $\partial \bar{D}_* / \partial y$), and (c) the difference $\Delta(\partial D / \partial y) = \partial \bar{D}_* / \partial y - \partial D / \partial y$, (d) histogram of global $\partial D/\partial y$, and (e) histogram of global $\partial \bar{D}_* / \partial y$.
