# Peer review of "Technical Note: Two types of absolute dynamic ocean topography"

_Ocean Science, 2018_

## Referee Comment (RC1) · Anonymous Referee #1 · 4 Jun 2018

I found this paper difficult to understand. After reading it several times, I think that it isn't really about two different oceanic dynamic topographies. It's really about two different ways to estimate a geoid, and thus the ocean sea surface height relative to that geoid. Is it not true that were data perfect, the two geoid estimates would necessarily agree? (The question is posed on line 119, but the answer uses two different data sets with differing error budgets and hence would never be identical. What does theory say?) There are also a number of confusing statements. Going through, roughly in page order,

I don't understand the first sentence of the Introduction. What is a spherical harmonic model in a flat-Earth approximation?

Eq. (1) is also applicable if defined relative to the center of the Earth..

V appears not to be the gravitational potential, but the anomaly of the gravitational potential.

If g is really fixed in Eq. 2, it should be written as g\_0 as e.g., in Eq. (6) \Delta g appears on the left of the equation and also on the right. Are we meant to interpret g as g=g+\Delta g?? Brun's formula requires a reference.

The paragraph starting on line 60 seems to be the crux of the issue, or of my misunderstanding. H\_r is called a "reference depth". Is this different from what is usually called the "level of no motion"? Is it not true that if the actual geostrophic velocity were known at this depth, it would be a "level of \*known\* motion" and the two definitions of D would coincide? The paper states that the differences between the two definitions of D is beyond the scope of the paper. But isn't this just the problem of classical oceanography of figuring out whether there is a deep reference level where the geostrophic velocity actually vanishes? Or at least getting an error estimate on it? If my reading of this paragraph is incorrect, then I do not understand at all what the paper is trying to say.

Going on, doesn't the traditional marine geoid really require that the water density should be constant? If it is constant, then it follows from the geostrophic relationship that no flow exists. But if the density is a function of depth, there is (to first approximation) no flow, but the integrated water density and hence absolute height would depend upon the vertical distribution.

The ocean circulation is forced by buoyancy and wind. Does the minimum energy argument apply to a forced system where energy put in must also be dissipated (line 284)? Maybe the corrections are negligible here?

Perhaps I have completely misunderstood this paper, but if so, it needs to be completely rewritten to make it less obscure, and to explain what it actually means.

---

## Author Comment (AC1) · 9 Jun 2018

I appreciate very much your comments, which are extremely useful for the revision. Below are my responses to these comments.

(1) "I found this paper difficult to understand. After reading it several times, I think that it isn't really about two different oceanic dynamic topographies. It's really about two different ways to estimate a geoid, and thus the ocean sea surface height relative to that geoid. Is it not true that were data perfect, the two geoid estimates would necessarily agree?"

I am very sorry for the confusion. I would like to clarify that the note is really about two different absolute dynamic ocean topographies (DOT) (that is why I submitted to

the Ocean Sciences). Since DOT (referred to the absolute DOT throughout the article) is the difference between the sea surface height (S) and geoid (N_hat) (see Eq(1)]. Thus, DOT depends on the use of geoid. Two-types of geoid estimates are of course different. They impact more on oceanography than on marine geodesy since both sea surface height and geoid are two-orders of magnitude higher than DOT [Wunsch and Gaposchkin, 1980; Bingham et al., 2008].

In Line-70, the classical marine geoid (i.e., the average level of SSH if the water is at rest) is defined as the first type geoid and denoted by N. In Line-103, the satellite determined marine geoid [Tapley et al., 2003; Shum et al., 2011] is defined as the second type marine geoid and denoted by N*.

The horizontal gradient of the first-type DOT represents the absolute surface geostrophic currents since no motion at the marine geoid N. See Eqs.(12)-(14).

The horizontal gradient of the second-type DOT does not represent the absolute surface geostrophic currents since water moves at the marine geoid N*.

I will make it clear in the beginning of the revised version that this note is addressing this problem.

(2) "(The question is posed on line 119, but the answer uses two different data sets with differing error budgets and hence would never be identical. What does theory say?)"

I apologize for the confusion. Only one data set for the second type MDOT(D_bar*) is used from the NASA/JPL website: https://grace.jpl.nasa.gov/data/get-data/dynamic-ocean-typography/.

The first-type DOT (D) is not observable.

It is the solution of Eq.(33) with the lateral boundary condition from [see Eq.(38)], the forcing term F calculated from the World Ocean Atlas 2013 (WOA13) temperature and salinity fields, which was downloaded from the NOAA National Centers for Environmental Information (NCEI) website:

https://www.nodc.noaa.gov/OC5/woa13/woa13data.html. (see Lines 212-215), and the bottom topography H downloaded from the NECI 5-Minute Gridded Global Relief Data Collection at the website: https://www.ngdc.noaa.gov/mgg/fliers/93mgg01.html. (see Lines 218-220).

The numerical solution of (D) is compared with MDOT(D_bar*).

(3) "I don't understand the first sentence of the Introduction. What is a spherical harmonic model in a flat-Earth approximation?"

I originally thought that this is the standard statement in marine geodesy's since I rephrased from a paper by Sandwell and Smith (JGR-Solid Earth, Vol 102. B5, Page 10,050, 1997) "The geoid height N(x) and other measurable quantities such as gravity anomaly $\Delta g(x)$ are related to the gravitational potential V(x, z) [Heiskanen and Moritz, 1967]. We assume that all of these quantities are deviations from a spherical harmonic reference Earth model so a flat-Earth approximation can be used for the gravity computation (A 10)."

I will change the statement in the revised version.

(4) "Eq. (1) is also applicable if defined relative to the center of the Earth"

I will add this in the revised version.

(5) "V appears not to be the gravitational potential, but the anomaly of the gravitational potential."

Yes. I will correct it in the revised version.

(6) "If g is really fixed in Eq. 2, it should be written as g0 as e.g., in Eq. (6)."

In Lines 44-45, I mentioned g is fixed constant in Eq.(2) as well as in Eq.(6) "where g = 9.81 m/s2, is the globally mean normal gravity, which is usually represented by g0 in geodesy."

I made this statement since the majority readers are oceanographers.

(7) "$\Delta g$ appears on the left of the equation and also on the right. Are we meant to interpret g as g=g+\Delta g??"

No. g in Eq(2), Eq(5), and Eq(6) is g0 in geodesy.

(8) "Brun's formula requires a reference."

I will add the reference for the Brun 's formula in the revised version.

(9) "The paragraph starting on line 60 seems to be the crux of the issue, or of my misunderstanding. Hr is called a "reference depth". Is this different from what is usually called the "level of no motion"? Is it not true that if the actual geostrophic velocity were known at this depth, it would be a "level of *known* motion" and the two definitions of D would coincide? The paper states that the differences between the two definitions of D is be- yond the scope of the paper. But isn't this just the problem of classical oceanography of figuring out whether there is a deep reference level where the geostrophic velocity actually vanishes? Or at least getting an error estimate on it? If my reading of this paragraph is incorrect, then I do not understand at all what the paper is trying to say."

This paragraph (Lines 60-67) may not be relevant and should be deleted in the revised version. My original purpose is to distinguish the absolute and relative DOTs since some oceanographers still use relative DOT. This note is only for the absolute DOT.

(10) "Going on, doesn't the traditional marine geoid really require that the water density should be constant? If it is constant, then it follows from the geostrophic relationship that no flow exists. But if the density is a function of depth, there is (to first approximation) no flow, but the integrated water density and hence absolute height would depend upon the vertical distribution."

The traditional marine geoid is the average level of SSH if the water is at rest. It does not require that the water density should be constant.

The traditional marine geoid is theoretically defined and hard to observe. The difference between SSH and the traditional marine geoid, i.e., the first-type absolute DOT can be determined by the ocean surface absolute geostrophic velocity, which is represented by its horizontal gradient (i.e., First-type absolute DOT).

(11) "The ocean circulation is forced by buoyancy and wind. Does the minimum energy argument apply to a forced system where energy put in must also be dissipated (line 284)? Maybe the corrections are negligible here?"

In a book chapter: "Veronis, G., Dynamics of large-scale ocean circulation, in Evolution of Physical Oceanography, B. A. Warren and C. Wunsch, eds., M.I.T. Press, Cambridge, MA, 140–184. 1980," George Veronis states that "... This general result (i.e., conservation of potential vorticity) for a dissipation-free fluid does not apply precisely to sea water where the density is a function not only of temperature and pressure but also of the dissolved salts. The effect of salinity on density is very important in the distribution of water properties. However, for most dynamic studies the effect of the extra state variable is not significant and (5.5) (i.e., the conservation of potential vorticity is valid." (see page 142 of that chapter).

Thus, the minimum energy argument for the geostrophic flow is basically valid.

(12) "Perhaps I have completely misunderstood this paper, but if so, it needs to be completely rewritten to make it less obscure, and to explain what it actually means."

I will rewrite the manuscript completely according to your as well as other reviewers' comments to make less obscure.

I am thinking to change the title into "Does satellite determined dynamic ocean topography represent the absolute surface geostrophic currents?"

Please also note the supplement to this comment:
https://www.ocean-sci-discuss.net/os-2018-51/os-2018-51-AC1-supplement.pdf

---

## Referee Comment (RC2) · C. K. Shum (Referee) · 10 Jul 2018

480

[referee-annotated manuscript omitted]

---

## Author Comment (AC2) · 28 Jul 2018

I appreciate very much your comments, which are extremely useful for the revision. Below are my responses to these comments. My responses to your comments are listed as follows with the line numbers in the responses are referring to the revised version.

(1) "I found this paper difficult to understand. After reading it several times, I think that it isn't really about two different oceanic dynamic topographies. It's really about two different ways to estimate a geoid, and thus the ocean sea surface height relative to that geoid. Is it not true that were data perfect, the two geoid estimates would necessarily agree?"

[Figure]

I am very sorry for the confusion.

I rewrote the Introduction section and made it clear that the note is really about two different absolute dynamic ocean topographies (DOT) (that is why I submitted to the Ocean Sciences). Since DOT (referred to the absolute DOT throughout the article) is the difference between the sea surface height (S) and geoid ( ) (see Eq(1)]. Thus, DOT depends on the use of geoid. Two-types of geoid estimates are of course different. They impact more on oceanography than on marine geodesy since both S and are two-orders of magnitude higher than DOT [Wunsch and Gaposchkin, 1980; Bingham et al., 2008].

In Lines 65-67, the classical marine geoid (i.e., the average level of SSH if the water is at rest) is defined as the first type geoid and denoted by N. In Lines 99-102, the satellite determined marine geoid [Tapley et al., 2003; Shum et al., 2011] is defined as the second type marine geoid and denoted by N*.

The horizontal gradient of the first-type DOT represents the absolute surface geostrophic currents since no motion at the marine geoid N. See Eqs.(12)-(14).

The horizontal gradient of the second-type DOT does not represent the absolute surface geostrophic currents since water moves at the marine geoid N*.

(2) "(The question is posed on line 119, but the answer uses two different data sets with differing error budgets and hence would never be identical. What does theory say?)"

I apologize for the confusion. Only one dataset for the second type MDOT( ) is used from the NASA/JPL website: https://grace.jpl.nasa.gov/data/get-data/dynamic-ocean-typography/.

The first-type DOT (D) is not observable.

It is the solution of Eq.(33) with the lateral boundary condition from [see Eq.(38)], the forcing term F calculated from the World Ocean Atlas 2013 (WOA13) temperature and salinity fields, which was downloaded from the

NOAA National Centers for Environmental Information (NCEI) website: https://www.nodc.noaa.gov/OC5/woa13/woa13data.html. (see Lines 216-219), and the bottom topography H downloaded from the NECI 5-Minute Gridded Global Relief Data Collection at the website: https://www.ngdc.noaa.gov/mgg/fliers/93mgg01.html (see Lines 222-224).

The numerical solution of (D) is compared with MDOT( ).

(3) "I don't understand the first sentence of the Introduction. What is a spherical harmonic model in a flat-Earth approximation?"

I originally thought that this is the standard statement in marine geodesy's since I rephrased from a paper by Sandwell and Smith (JGR-Solid Earth, Vol 102. B5, Page 10,050, 1997) "The geoid height N(x) and other measurable quantities such as gravity anomaly $\triangle$g(x) are related to the gravitational potential V(x, z) [Heiskanena nd Moritz, 1967]. We assume that all of these quantities are deviations from a spherical harmonic reference Earth model so a flat-Earth approximation can be used for the gravity computation (A 10)."

I revised it into standard oceanographic expression for coordinate system: "Let the coordinates (x, y, z) be in zonal, latitudinal, and vertical directions" (see Line-38).

(4) "Eq. (1) is also applicable if defined relative to the center of the Earth"

I added this statement in the revised version (see Lines 42-43).

(5) "V appears not to be the gravitational potential, but the anomaly of the gravitational potential."

I corrected (see Line 45).

(6) "If g is really fixed in Eq. 2, it should be written as g0 as e.g., in Eq. (6)."

In Lines 47-48, I mentioned g is a fixed constant in Eq.(2) as well as in Eq.(6) → "where g = 9.81 m/s2, is the globally mean normal gravity, which is usually represented by g0

in geodesy."

I made this statement since the majority readers are oceanographers.

(7) "$\Delta g$ appears on the left of the equation and also on the right. Are we meant to interpret g as g=g+\Delta g??"

No. g in Eq(2), Eq(5), and Eq(6) is g0 in geodesy.

(8) "Brun's formula requires a reference."

I added the reference for the Brun 's formula in the revised version (see Line 46).

(9) "The paragraph starting on line 60 seems to be the crux of the issue, or of my mis-understanding. Hr is called a "reference depth". Is this different from what is usually called the "level of no motion"? Is it not true that if the actual geostrophic velocity were known at this depth, it would be a "level of *known* motion" and the two definitions of D would coincide? The paper states that the differences between the two definitions of D is be- yond the scope of the paper. But isn't this just the problem of classical oceanography of figuring out whether there is a deep reference level where the geostrophic velocity actually vanishes? Or at least getting an error estimate on it? If my reading of this paragraph is incorrect, then I do not understand at all what the paper is trying to say."

This paragraph (Lines 60-67 in the original version) was not relevant and deleted.

(10) "Going on, doesn't the traditional marine geoid really require that the water density should be constant? If it is constant, then it follows from the geostrophic relationship that no flow exists. But if the density is a function of depth, there is (to first approximation) no flow, but the integrated water density and hence absolute height would depend upon the vertical distribution."

The traditional marine geoid is the average level of SSH if the water is at rest. It does not require that the water density should be constant. The traditional marine

geoid is theoretically defined and hard to observe. The difference between SSH and the traditional marine geoid, i.e., the first-type absolute DOT can be determined by the ocean surface absolute geostrophic velocity, which is represented by its horizontal gradient (i.e., First-type absolute DOT).

(11) "The ocean circulation is forced by buoyancy and wind. Does the minimum energy argument apply to a forced system where energy put in must also be dissipated (line 284)? Maybe the corrections are negligible here?"

In a book chapter: "Veronis, G., Dynamics of large-scale ocean circulation, in Evolution of Physical Oceanography, B. A. Warren and C. Wunsch, eds., M.I.T. Press, Cambridge, MA, 140–184. 1980," George Veronis states that "... This general result (i.e., conservation of potential vorticity) for a dissipation-free fluid does not apply precisely to sea water where the density is a function not only of temperature and pressure but also of the dissolved salts. The effect of salinity on density is very important in the distribution of water properties. However, for most dynamic studies the effect of the extra state variable is not significant and (5.5) (i.e., the conservation of potential vorticity is valid." (see page 142 of that chapter).

Thus, the minimum energy argument for the geostrophic flow is basically valid.

(12) "Perhaps I have completely misunderstood this paper, but if so, it needs to be completely rewritten to make it less obscure, and to explain what it actually means."

I rewrote the manuscript completely according to your as well as the second reviewer 's comments to make less obscure.

Please also note the supplement to this comment:
https://www.ocean-sci-discuss.net/os-2018-51/os-2018-51-AC2-supplement.pdf

---

## Author Comment (AC3) · 28 Jul 2018

Reviewer #2 (Prof. C.K. Shun) 's outstanding review and encouragement are highly appreciated. The comments are extremely useful for the revision. Below are my responses to these comments.

Assessment:

" . . . As a result, the author concluded more studies need to be done based on the finding which indicated that "the satellite determined DOT does not conform with the basic physical oceanography principle of geostrophic currents". While this original study may be unconventional, but the hypothesis stated and the approach based on the first principle to reveal the differences of the two types of DOTs commonly used is novel,

[Figure]

I recommend publications with minor revisions. See the attached annotation of the manuscript."

Thank you very much for your encouragement and support.

Annotated Comments in Supplement

The annotated comments are extremely important. My responses are listed as follows.

(1) "The terms 'geoid' (or 'marine geoid') and DOT (or MDOT) in the text, have been used interchangeably, causing a bit of confusion. Please check and make sure that there were not errors."

Done.

(2) "DOTs are NOT equal to marine geoid? please clarify the question. It is apparent that you are contrasting two types of DOTs (not contrasting DOTs and geoids?). May be you mean AND the differences of the two types of geoids. Please make it more clear (Line 119 in original version)."

I revised: "Do the horizontal gradients of the second type MDOT ( ) represent the absolute surface geostrophic currents?" See Lines 117-118.

(3) Marked editorial corrections

Done.

Please also note the supplement to this comment:
https://www.ocean-sci-discuss.net/os-2018-51/os-2018-51-AC3-supplement.pdf

[Figure]

**Supplement:**

[revised manuscript text omitted]